# Beam Search Algorithm for Ship Anti-Collision Trajectory Planning

**DOI:** 10.3390/s19245338

**Published:** 2019-12-04

**Authors:** Joanna Karbowska-Chilinska, Jolanta Koszelew, Krzysztof Ostrowski, Piotr Kuczynski, Eric Kulbiej, Piotr Wolejsza

**Affiliations:** 1Faculty of Computer Science, Bialystok University of Technology, 15-351 Bialystok, Poland; j.karbowska@pb.edu.pl (J.K.-C.); j.koszelew@pb.edu.pl (J.K.); k.ostrowski@pb.edu.pl (K.O.); 2Unity Developer, The Dust; 50-043 Wrocław, Poland; keluthaz@gmail.com; 3R&D Department, Sup4Nav sp. z o.o.; 71-602 Szczecin, Poland; eric.kulbiej@sup4nav.com; 4Faculty of Computer Science and Telecommunication, Maritime University of Szczecin, 70-500 Szczecin, Poland

**Keywords:** anti-collision trajectories, navigational decision support system, autonomous ship, beam search algorithm (BSA)

## Abstract

The biggest challenges in the maritime environment are accidents and excessive fuel consumption. In order to improve the safety of navigation at sea and to reduce fuel consumption, the strategy of anti-collision, shortest trajectory planning is proposed. The strategy described in this paper is based on the beam search method. The beam search algorithm (BSA) takes into account many safe trajectories for the present ship and chooses the best in terms of length and other criteria. The risk of collision of present ship with any target ships is detected when the closest point of approach (CPA) of the present ship is violated by the target ship’s planned trajectory. Only course alteration of the present ship is applied, and not speed alteration. The algorithm has been implemented in the decision support system NAVDEC and tested in a real navigation environment on the m/f Wolin, a Polish ferry. Almost all BSA trajectories calculated were shorter in comparison to the standard NAVDEC-calculated algorithm.

## 1. Introduction

The shipping industry is one of the most important sectors from an economic point of view, because of its continuous growth and 90% share of world trade. On the other hand, the increasing density of ships increases the number of collisions.

Modern ships are fitted with modern navigational devices that provide a large amount of information and real-time data. The automatic identification system (AIS) transmits information, including the position of ships, their course, speed, and identity. The automatic radar plotting aid (ARPA) calculates target ships’ courses and speeds and transmits information about a risk of collision. The electronic chart display and information system (ECDIS) integrates on one display the electronic chart and information from navigational sensors. However, the above information systems do not offer the navigator ready anti-collision solutions when there are ships approaching on a collision course. Despite the modern equipment on ships, there are a large number of collisions every year. The European Maritime Safety Agency has reported that human error was responsible for 85 per cent of accidental events in the year 2018 [1]. Commonly used navigational systems on ships mostly perform the information function. It is the navigator who analyzes a large amount of data from the devices on the ship and decides what maneuver to perform in collision situations. From an economic point of view, the ship’s trajectory should be safe, but, at the same time, the shortest trajectory, in order to minimize fuel consumption. A medium-size container vessel (8000 TEU) typically burns 260 tons of fuel every day if proceeding at ‘full ahead’. This means that for every minute, 170 L of fuel is used. If an anti-collision maneuver is shortened by one minute only, over 62 tons of fuel can be saved every year (assuming that such a ship executes only one maneuver per day). To avoid deviation from the planned trajectory, it is possible to reduce/increase speed. However, from the perspective of safety, course alteration is considered to be faster and more visible for others, especially for radar observers. Besides, weather-optimized routing is an efficient tool for the overall voyage plan to save fuel. Route planning is usually divided into two parts. The strategic part is related to weather-optimized routing. The tactical part is connected with collision avoidance. The tactical part is executed as an element of the strategic part. The experiment presented in the article is focused on the tactical part of route planning.

In recent years, in the literature, there have been a number of approaches to generating anti-collision and optimal trajectories for ships [2,3]. Most of the presented methods consider two ships or multi-ship encounter situations, in compliance with the COLREGs (Collision Regulations) rules [4] or good seamanship. Some of these methods could be part of a navigational decision support system [5,6,7], which could help a navigator to analyze the data from sensors and propose a set of safe and economic maneuvers in a collision situation.

Existing deterministic methods of generating anti-collision trajectories converge to the best solution, but in more complex cases at sea, these methods are sometimes unable to indicate an anti-collision path because of high computational time. The cooperative path planning algorithm [8] computes paths for all ships involved in the encounter in a cooperative mode. In the trajectory-based algorithm [7], a database of trajectories is searched to find the shortest trajectory which will not collide with the other and meet COLREGs rules.

The deterministic method is also used in the navigation decision support system called NAVDEC [6]. Anti-collision maneuvers are determined analytically for the present ship by the system, taking into account COLREGs rules and good seamanship. Certified by the Polish Register of Shipping, the NAVDEC, as commercialized software, is used on a number of ships.

Many studies have investigated the ship path planning problem as a multi-objective optimization problem [9] which can be solved in real-time. Heuristic methods, such as genetic algorithms, ant colony algorithms, and particle swarm optimization have been used. In [10], an evolutionary algorithm was developed, producing a near-optimal set of safe paths, fulfilling COLREGs rules, for all ships engaged in a collision situation. Tsou [11] presented a description of decision support systems which utilize AIS and ECDIS real data for the detection of a collision risk. The evolutionary computation was used by Tsou to generate the optimal anti-collision trajectory. An ant colony algorithm was implemented in the decision support system for ship route planning by Tsou [12] and Lazarowska [13]. The particle swarm optimization was also used by Kang [14] to create an anti-collision trajectory under COLREGs rules in a navigation environment. Other research approaches include artificial neural networks [15], cooperative multi-person positional modelling games [16], artificial potential fields [17], or the visibility graph method [18]. In [19], the fast marching method is used in a multilayer path planner for an unmanned surface vehicle. This planner takes into account complex marine environments with coastal and dynamic obstacles. In [17,20], the artificial potential field method is presented as a tool to solve the local optimization of path planning and to avoid dynamic obstacles. It can be noted that the domain of ships is an important approach to deal with the optimal path planning problem [21,22].

The review of the above research indicates that the problem of generating anti-collision and optimal trajectories in maritime navigation is an open research problem. On the other hand, only a few of the proposed algorithms have been tested in real conditions as a component of a decision support system [6,15]. Most studies do not describe how these algorithms work in the case when the present ship is located in dense traffic waters. In the analyzed algorithms, the COLREGs are usually followed, but, in practice, this is not always possible (e.g., in restricted visibility conditions, most vessels do not fulfil “safe speed” requirements because it results in huge delays and interruptions of port and vessel schedules).

This study examines the concept of the beam search algorithm (BSA) [23], adopted to obtain the shortest anti-collision trajectory. The proposed BSA algorithm generates many safe trajectories for the present ship and chooses the best, in terms of length. The risk of collision between the present ship and any of the target ships is detected when the closest point of approach (CPA) of the present ship is crossed by a target ship’s planned trajectory. The algorithm has been included in the decision support system NAVDEC [24,25] and tested in a real navigation environment on the m/f Wolin, which travels daily on the Świnoujście-Trelleborg route. The lengths of anti-collision trajectories generated by the standard NAVDEC algorithm and BSA have been compared. Almost all BSA trajectories have been shorter than those calculated from the standard NAVDEC algorithm.

The authors consider anti-collision systems [26] as critical elements of autonomous navigation. Despite legislation challenges, which definitely slow down the introduction of autonomous ships into international shipping, there are ongoing projects which aim to build autonomous or semi-autonomous coasters. They tend to sail on national waters only, e.g., Yara Birkeland in Norway, and thus, they do not face international legislation challenges.

A wider vision of autonomous shipping was presented in the following papers [27,28,29,30] while the algorithms presented in this paper will be implemented on a semi-autonomous coaster (SAC) [31], a project executed by the consortium led by the Szczecin Shipyard with the delivery due in 2021.

## 2. Generating of Anti-Collision Trajectories for a Ship

The authors propose the BSA algorithm, based on a beam search strategy [23], to determine a set of anti-collision trajectories for a present ship. The algorithm explores a tree of trajectories (formed by risk-avoiding left and right turns by the present ship), and at each step it stores a group of the most promising partial solutions, which are expanded by adding subsequent maneuvers. The algorithm has the following inputs:Planned itineraries of the present ship and *N* target ships. Each itinerary is determined by successive waypoints and can be treated as a polygonal chain (segments between successive waypoints).Speeds and courses of all ships (it is assumed that ships do not alter their speed, only their course).Minimal *D_CPA_* (Distance at the closest point of approach) feasible for the present ship. The *D_CPA_* concept is explained in the text below and in Figure 1.The maneuverability of the present ship, namely, the maximal angle of turn β and minimal distance between subsequent maneuvers *d*. Both parameters are ship-dependent, and their values are set in such a way that the present ship does not reduce considerably reduce speed in subsequent maneuvers (i.e., *d* is usually equal to 4–5 lengths of the ship).

In Figure 1 the numbers by the trajectories indicate the ships’ positions after subsequent time steps. When t = 5, the ships are closest to each other. The distance at the closest point of approach (*D_CPA_*) is equal to 106 here (lattice in the figure has a size of 100).

To assess a collision risk between the present ship and any target ship, the algorithm uses the *D_CPA_* and *T_CPA_* measures. *D_CPA_* is the closest distance between two ships, assuming that both maintain their current courses and speeds, while *T_CPA_* is the time needed to reach *D_CPA_*. The formulas are given below:(1)$$D_{CPA}=R\cdot sin\alpha$$
where *R* is the distance between two vessels and *α* is the angle between the vector of relative speed $\left(\underset{V_{own}}{\to }-\underset{V_{target}}{\to }\right)$ and the vector of relative position $\left(\underset{P_{target}}{\to }-\underset{P_{own}}{\to }\right)$ If *α* > 90 degrees, then *D_CPA_* = *R*.(2)$$T_{CPA}=\frac{-XV_{RX}+YV_{RY}}{V_{R}^{2}}$$
where *X* and *Y* denote the present relative position of the target, *Vrx* and *Vry* denote the components of target relative velocity, *Vr* is the relative speed, and *T_CPA_* is the time to reach the closest point of approach [32].

The BSA starts with the trajectory of the present ship (i.e., the planned itinerary). Then, it checks whether the currently analyzed segment of the trajectory faces a collision risk (i.e., *D_CPA_* < minimal *D_CPA_*) with any target ship, known as Target(i).

If such a situation arises, two new trajectories are formed (based on the current trajectory segment). One of them includes a starboard turn of minimal angle and the other includes a port side turn of minimal angle, in order to pass the target ship, Target(i), at a distance of at least minimal *D_CPA_*. After passing Target(i) safely, another maneuver is added to the created trajectory in order to turn back towards the next planned waypoint. The point of turning back is calculated using a binary search, where it is the closest point where turning towards the planned waypoint will not violate minimal the *D_CPA_* constraint with respect to ship Target(i). If the first created segment (which avoids a collision) is safe with respect to the other target ships, then the processed trajectory is added to the set of current solutions (the second segment, which goes back to the waypoint, will be analyzed in another iteration). If the final waypoint is reached safely, the currently processed trajectory is added to the set of final solutions.

If the present ship is safe with respect to target ship Target(i), the algorithm creates an artificial collision course and two anti-collision trajectories, similar to those described in the previous paragraph. This helps to explore the solution space in a more effective way. In some situations, only one maneuver is enough to pass a group of ships.

At each step, the algorithm stores not more than maxN best currently built trajectories (in terms of expected length) and expands them by adding new maneuvers, which enables the present ship to pass subsequent target ships at a safe distance. Afterwards, maxN best newly-created trajectories are chosen to be processed in the next step. Thus, the algorithm can be classified as a beam search, which explores the tree of safe trajectories (built by port and starboard turns) and at each step expands maxN with the most promising partial solutions. The pseudocode of the Algorithm 1 is given below:
**Algorithm 1:** Beam Search1: CT: List of current trajectories, where T is one trajectory in the list (CT is initially empty)2: FT: Set of final trajectories (FT is initially empty)3: OT: Own trajectory.4: asT: A current processed segment of T trajectory.5: maxN: Maximal number of processed solutions.6: Add OT to CT7: **while** CT is not empty **do**8:   Remove T (the first trajectory of CT) from the start of list CT9:   **for each** Target(i) ship’s trajectory **do**10:     **if** the *D_CPA_* of the present ship is violated in section asT by Target(i)’s trajectory **then**
11:       Generate two anti-collision maneuvers (port and starboard) and segments back to the waypoint12:       Based on the two maneuvers and T, create two new trajectories (T_1_ and T_2_)13:     **else**14:       Generate an artificial collision course T_A_ of the present ship with Target(i)’s trajectory.15:       For the artificial course T_A_, generate two anti-collision maneuvers (port and starboard) and segments back to the waypoint16:       Based on the two maneuvers and T_A_, create two new trajectories (T_1_ and T_2_)17:     **end if**18:     **for each** trajectory from {T_1_, T_2_} set **do**19:       **if** the first of two newly created segments of Ti trajectory is *D_CPA_* safe with respect to all remaining ships **then**20:         **if** the segments are last segments in Ti and both are *D_CPA_* safe **then**21:           Add Ti to FT22:         **else**23:           Add Ti to the end of list CT24:         **end if**25:       **end if**26:     **end for**27:   **end for**28:   **if** size(CT) > maxN **then**29:     Remove size(CT) from maxN worst trajectories (in terms of expected length) from CT30:   **end if**31: **end while**32: Choose the best trajectory (or trajectories) from FT.

The maximum number of vessels which can be analyzed by the BSA is equal to 30. Example situations which illustrate the algorithm concept are given below. In Figure 2, present ship A (coordinates (0, 0)) is heading north towards the waypoint (large red circle). Its trajectory is marked in orange. However, it is on a collision course with target ship B (coordinates (200, 200), blue trajectory, western course). The intersection of the blue and orange lines marks the point of collision. Both ships have the same speed (50 units) and their positions after subsequent time steps are marked with dots and numbers (collision takes place at t = 4).

The BSA was used and computed two anti-collision trajectories for the present ship (marked in red). The minimal safe *D_CPA_* was 100 units (scale in the image is also equal to 100, for convenience). The trajectory, which requires a starboard turn followed by a port turn (and passes astern of the target ship), is more efficient, as it takes about 11 units of time to get to the waypoint (with a length of 550). The points of the closest approach are connected with a thin grey line (*D_CPA_* = 100, *T_CPA_* equal to around 3). The other trajectory takes almost 14 units of time (with a length of 700), so it is less efficient than the previous one.

Another situation is given in Figure 3. This time, present ship A has a collision risk with target ships B and C, which are approaching from opposite directions (blue lines), perpendicular to the trajectory of ship A (orange). The present ship is on a collision course with B and will pass too close to the ship C. The algorithm was used once again and generated an anti-collision trajectory for the present ship, which avoids both ships, passing at a safe distance (*D_CPA_* = 100). It consists of 3 segments (points of turns are marked with red circles). The trajectory length is about 575 units (the straight-line collision trajectory has a length of 500).

Another situation (Figure 4) involves three target ships. Ships B and C are fast (100 units of speed), while ship D has the same speed as A (50 units). Only the resulting trajectory is illustrated. At first, the present ship passes astern of ship B at a safe distance (starboard and port side turns). Another slight port turn enables the present ship to pass astern of ship C at a safe distance. Afterwards, the present ship cruises ahead of ship D and turns towards the waypoint. The generated trajectory has a total length of about 650 and is only slightly longer than the straight-line collision course (600 units).

## 3. Results

In this chapter we present the results of tests carried out on the m/f Wolin, conducted in real conditions, comparing two algorithms, namely, the BSA and the solution implemented in the NAVDEC system. The aim of the test was to calculate the difference between the length of the trajectory in both anti-collision algorithms while maintaining the superior condition of maintaining the CPA.

We ran the BSA during the tests using the following assumptions:The highest priority is given to safety, i.e., *D_CPA_* selected by the navigator should be kept at a maximum at all times.When the above condition is fulfilled, then the shortest route is chosen.Multi-stage (many course alterations to reach the final destination or the next waypoint) solutions are available.The route is updated each time after receiving a new AIS/ARPA message.COLREGs are not followed. The present vessel has the challenge of guessing whether other ships observe the COLREGs or not.The next waypoint is 15 Nm from the current position. A representative circle around the present ship is displayed in Figure 5, Figure 6, Figure 7 and Figure 8.The specific parameter of the BSA (denoted by maxN) means the maximal number of partial solutions taken into consideration on the tree structures generated in the algorithm. This parameter has a crucial impact on the quality of the result and the complexity of the algorithm. The higher value of maxN parameter, the closer the solution is to the optimum, at the expense of longer computation time. During the tests, maxN = 1000.The BSA run time was on average less than 0.1 s. The algorithm was executed every time when data were updated (each AIS message, at least every 10 s). To avoid frequent maneuvers, the algorithm tried to maintain the current trajectory (or make only small changes), unless the current trajectory was not CPA-safe anymore, or a much better trajectory was generated based on the updated locations.

General assumptions for the NAVDEC:The highest priority is given to safety, i.e., *D_CPA_* selected by the navigator should be kept at a maximum at all times.Only when the above condition is fulfilled, then the present vessel can proceed to the waypoint, where *T_CPA_* with all targets is nonpositive.Multi-stage solutions are not to be undertaken.The safety course is updated each time a new AIS/ARPA message is received.COLREGs are implemented in relation to a one-on-one encounter situation.Next waypoint is 15 Nm from the current position.

General assumptions during the testing period of the 30th of November to the 1st of December, 2018:*D_CPA_* = 1.0 Nm.The minimum distance between waypoints is 4 vessel lengths.The maximum course change in degrees is 75 degrees.

In all cases, the NAVDEC observed the COLREGs, which means that it always suggested an alteration to starboard. Some of these alterations had to be significant to in order fulfil the regulations. It was proven by the authors (Figure 5) that, in some cases, a small alteration to port could result in a significant reduction of the distance, particularly when there were many ships around. The case in Figure 5 can be considered as a heavy traffic situation in the Baltic Sea. Within six Nm from the Wolin, there were seven other vessels. At least one of them was on the opposite course, which made a speed reduction a non-effective maneuver. In this case, speed reduction could increase only *T_CPA_* while *D_CPA_* would remain the same when the other vessel kept her course and speed. Speed reduction is an effective maneuver in crossing situations and in restricted waters when course alteration is limited, for example, by the width of the channel. Such a maneuver reduces fuel consumption. However, in the open sea, course alteration is used much more often than speed reduction. Such instruction is issued by Masters in ‘standing orders’. Course alteration, where possible, is considered to be faster and more visible by other navigators. All the cases are presented in Table 1. Selected cases are presented in Figure 5, Figure 6, Figure 7, Figure 8 and Figure 9. Trajectories calculated by the BSA are drawn in red, while NAVDEC trajectories are marked in blue. The comparison, separately, for each case, is presented in the bottom right corner (the application solution).

In the situation presented in Figure 5, the advantage of the multi-stage BSA maneuver is well visible. BSA manages to calculate a safe trajectory in a dense traffic situation. Also, the navigator selected this solution even if it required a turn to port at the first stage. An alteration to starboard (NAVDEC) had to be significantly bigger.

Additionally, the one-stage maneuver was determined, in which the vessel had to go a long way. That solution was not accepted by the navigator as it was over 150% longer.

In the situation shown in Figure 6, both algorithms decided to plot a starboard route, which gave space to the approaching vessel (MMSI 245170000). However, the BSA found a much shorter route, which enabled the ship to save half a mile from the original 15 Nm distance (over a 3% reduction).

In the situation presented in Figure 7, both algorithms calculated almost the same initial course. However, the BSA suggested an earlier return to the original trajectory, which allowed the ship to save 0.14 Nm on the 15 Nm track (almost a 1% reduction), even if the initial deviation from the original course was bigger.

Almost the same situation is presented in Figure 8. In this case, the initial course suggested by both algorithms was the same, which was 336.6°. While the NAVDEC starts the returning course when *T_CPA_* to all objects is below zero, the BSA starts returning astern of the last dangerous object, which saves almost 1% of the track, i.e., 0.13 Nm.

In the case presented in Figure 9. The difference between the two safe solutions is substantial, reaching over a 12% reduction. The NAVDEC system decided to pass astern of all four targets involved in the collision situation, while the BSA, immediately after passing vessel Fredo, returned to the original trajectory, passing ahead of two targets at the presumed safe distance.

## 4. Discussion

The above discussed algorithms were tested in real conditions on the m/f Wolin. During one Świnoujście-Trelleborg-Świnoujście round trip, which lasts 15 h and covers over 200 Nm, the m/f Wolin had 14 collision situations. In all those collision situations, both algorithms calculated safe trajectories, which enabled the ship to pass all targets at the required *D_CPA_*. The total route generated by the BSA was 204.34 Nm, while that calculated by the NAVDEC was 280.84 Nm, which is 37% longer than the route calculated by the BSA. Referring the results to the container ship presented in the introduction, this could represent a saving of over 800 tons of fuel every year (14 collision avoidance maneuvers per day). The average global price of one ton of heavy fuel oil on the 13th of November, 2019, was 406.50 USD, which translates into a saving over 300,000 USD, and much more from an environmental protection perspective. This saving can almost be doubled when ships use marine gas oil, which is already required in certain areas around the world.

The average execution time of the single BSA call was less than 0.1 of a second. The BSA algorithm is called about every 10 s, as current data from AIS are downloaded every 10 s. The referenced value (2 s) mentioned in the introduction concerns the Lazarowska algorithm [7]. In the experiment, the phase 1 limit was set as 8 Nm. This means that vessels further than 8 Nm are not taken into account when calculating a collision avoidance maneuver. In a theoretical situation, when two vessels are in a head-on situation and both are proceeding at 25 knots, the distance of 8 Nm will be covered within 9 min. This is probably the shortest available time to avoid a collision.

The NAVDEC fulfils the COLREGs, while the BSA does not. Nevertheless, the ship using the BSA was able to pass all other targets at the assumed *D_CPA_*. At the same time, the executed trajectory was relatively shorter than that generated by the NAVDEC. In the authors’ opinion, it is worth considering using such tools, which fulfil the safety criteria, i.e., maintaining a safe distance, and enable the potential to reduce fuel consumption. Total savings on a global scale with this algorithm can be relatively high when considering the COLREGs, which force vessels to alter course to starboard when they are a give-way vessel. Even a small deviation to port can be treated as a violation of the COLREGs. This is why some of the solutions, which are the shortest, do not comply with the COLREGs. The goal of this paper is to show that it is possible to fulfil the safe distance criteria and make the route significantly shorter. Only two solutions advised an alteration to port, but that presented in Figure 9 cannot be treated as a COLREG violation. In the head-on situation presented there, when the *T_CPA_* is shorter than 5 min, altering course to starboard will lead to a very dangerous situation. Even the NAVDEC suggested an alteration to port, as it was difficult or even impossible to maintain a safe distance when turning to starboard. Summarizing the 14 cases, there was one violation. Even if it was excluded from the results, there is still a significant distance saving in comparison to the NAVDEC. Discussions on modifying the COLREGs have been in place for many years [31,33,34,35]. They will probably be intensified facing autonomous vessel challenge. Whilst the COLREGs and other regulations may be seen as a firm set of rules, they also provide for the exercise of human judgment. For example, under Rule 2b included in the COLREGs, a seafarer may depart from the rules to avoid an incident in the case of a great risk of a collision where an alteration to starboard side and speed reduction cannot stop a collision which is certain. It is one of many challenges in autonomous shipping. It is true that it is still the issue of the future, but, undoubtedly, some of the rules, which were set forth in the early 1970s, need to be revised.

These authors are aware that the presented solution is not in line with the regulations such as the COLREGs. However, it is easy to change the BSA so that, for example, the first alteration of the course is to starboard. In this situation, the probability that the generated trajectory is optimal in terms of length will be decreased. It is worth emphasizing that the safe *D_CPA_* during the entire length of the anti-collision trajectory is the most important and inviolable condition of the BSA method. It is only secondly that the condition of the shortest possible length of the trajectory is taken into account.

The results were also affected by other assumptions than those mentioned above. The distance between two waypoints does not allow for frequent course alterations, which are practically not feasible for the vessel. The assumption of the maximum course alteration also excluded some possible solutions. This was done on purpose, as significant course alterations significantly reduce vessel speed. Even a small alteration has an influence on speed. In the case of m/f Wolin, changes by 30 degrees do not influence the passage time. They reduce the speed, but, at the same time, the route is also shortened, as the vessel does not precisely pass the waypoint, starting the turn before reaching the waypoint and reaching a new leg after the waypoint. Ultimately, the shortcut makes speed reduction not important.

## 5. Conclusions

The article presents an original heuristic BSA algorithm that generates an anti-collision trajectory, which can be composed of at least two sections of a polygonal line, ensuring the safety condition, which is the minimal value of *D_CPA_*, set for the entire length of the suggested trajectory. The algorithm also seeks to choose the shortest possible trajectory while maintaining *D_CPA_* preservation. The resultant trajectory is determined on the basis of the navigation data from the perspective of the given ship and the present ship. It is very important that the BSA can be applied not only to a small number of ships involved in a collision situation (as in [7]). The maximum number of vessels which can be analyzed by the BSA is 30, which allows for the use of our method not only in open waters, but also in restricted areas. Another important feature of the BSA is that this method takes into account all kinds of collision situations, including the most dangerous ones, such as the last moment maneuver [26].

Directions for further research have already been closely determined. One concern is the addition of dynamic ship aspects, which depends on ship size and maneuverability, considering that dynamics trajectories will reflect real conditions. An interesting approach regarding ship dynamics is presented in [36,37], where the trajectory-generation problem is divided into separate stages, where ship dynamics are solved in the last stage. Our further research will concentrate on a way to incorporate ship dynamics in the BSA (as a separate stage or an integral part of the BSA).

Another direction of our research is more complicated, but also beneficial. The BSA structure allows for the application of the main idea of the algorithm to solve the general problem of generating a set of safe and optimal trajectories for the vessels involved, rather than the present ship alone. In this approach, we have the navigation data of a number of ships as the input in the given sea area (for example, with a radius of 30 Nm). The objective of the solution is to generate an entire set of anti-collision trajectories which do not violate *D_CPA_* for each pair of vessels and have the lowest possible sum of the trajectory lengths. This more general version of the problem is very important for the development of autonomous ship technologies. When it comes to mass deployment, software design for remote decision-making centers will be very important. Such solutions will generate and directly transmit a set of trajectories for the whole configuration of vessels detected in the given area to the ship autopilots, instead of a single anti-collision trajectory.

## Figures and Tables

**Figure 1 sensors-19-05338-f001:**
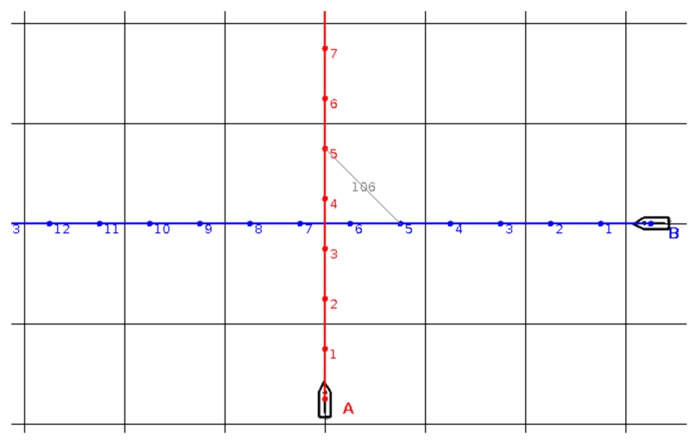
Trajectories of two ships: Ship A heading north and ship B heading west at the same speed.

**Figure 2 sensors-19-05338-f002:**
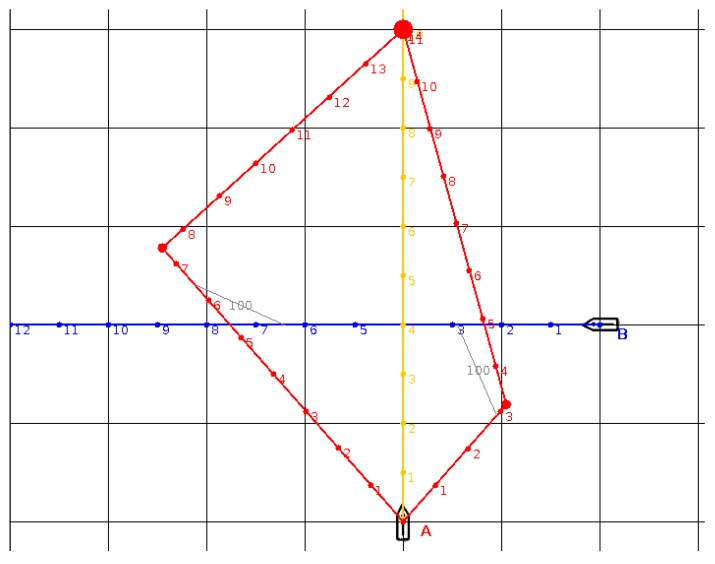
The situation with one B target ship. The initial trajectory of own ship A is orange. The trajectory of target ship B is marked in blue. Two anti-collision trajectories for A ship, proposed from BSA algorithm, are marked in red.

**Figure 3 sensors-19-05338-f003:**
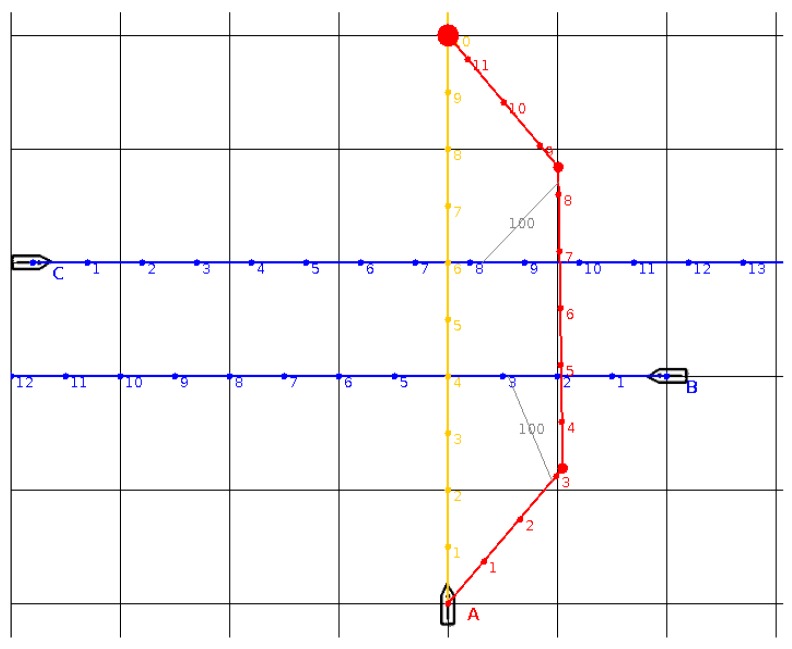
The situation with two B and C target ships. The initial trajectory of own ship A is orange. The trajectories of target ships B and C are marked in blue. The anti-collision trajectory for the A ship, proposed from BSA algorithm, is marked in red.

**Figure 4 sensors-19-05338-f004:**
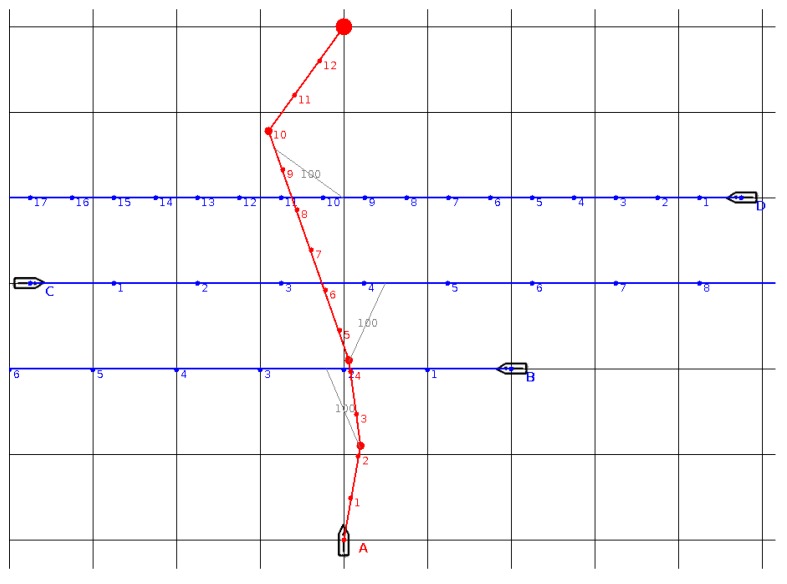
The situation with three B, C, D target ships (the trajectories of the ships are blue). The anti-collision trajectory for the A ship, proposed from BSA algorithm, is marked in red.

**Figure 5 sensors-19-05338-f005:**
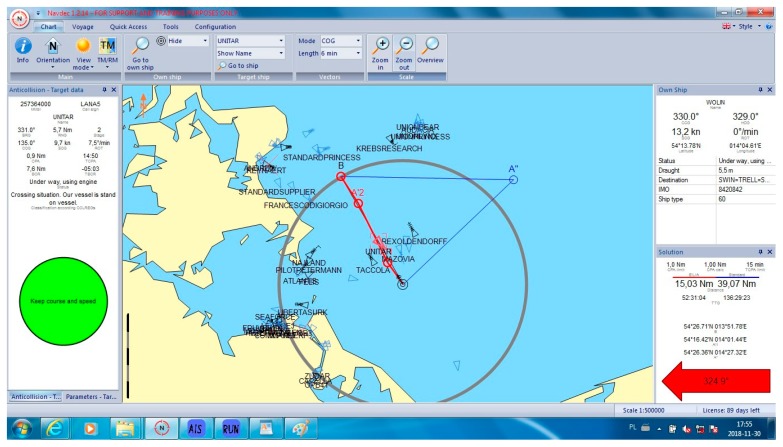
Collision situation number 3 from Table 1.

**Figure 6 sensors-19-05338-f006:**
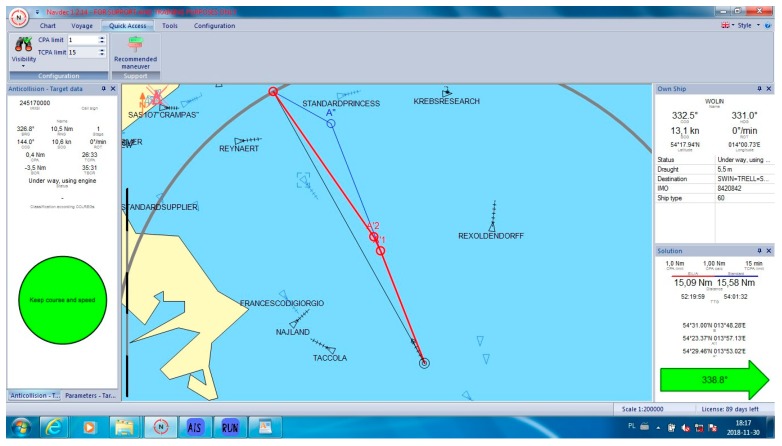
Collision situation number from Table 1.

**Figure 7 sensors-19-05338-f007:**
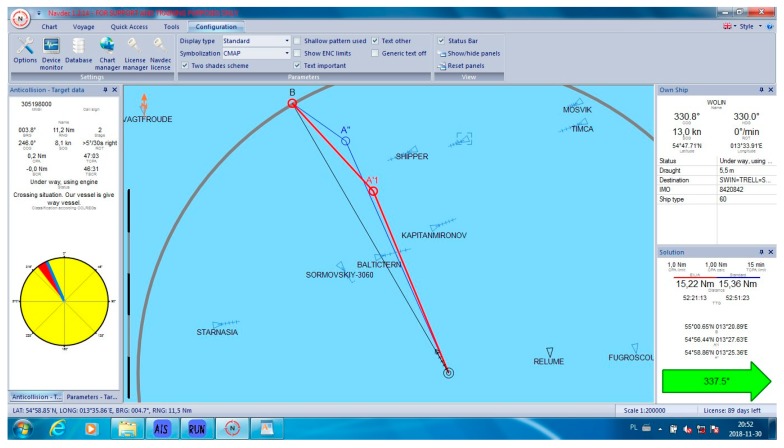
Collision situation number 8 from Table 1.

**Figure 8 sensors-19-05338-f008:**
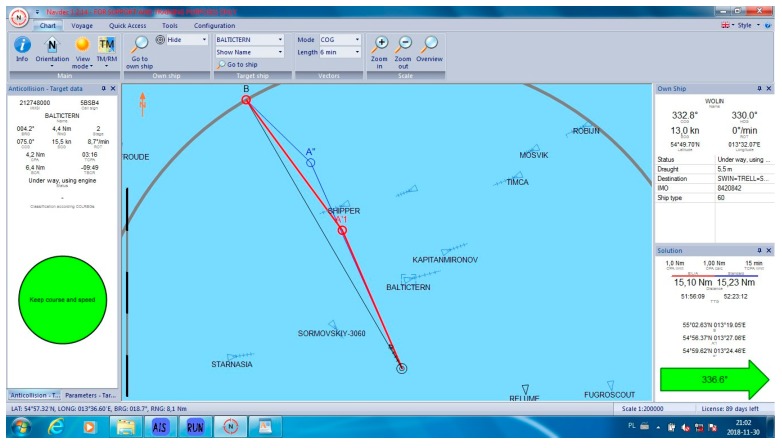
Collision situation number from Table 1.

**Figure 9 sensors-19-05338-f009:**
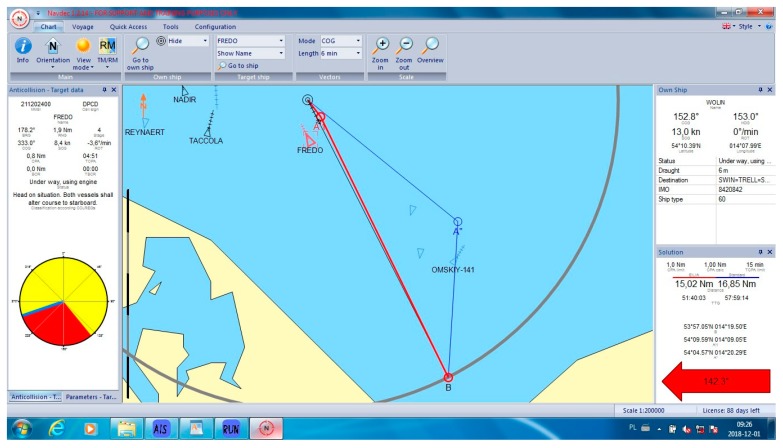
Collision situation number 14 from Table 1.

**Table 1 sensors-19-05338-t001:** Summary of all collision situations during the 30 November to the 1 December 2018, voyage.

Collision Situation	BSA Initial Course (°)	NAVDEC Initial Course (°)	BSA Trajectory to the Next Waypoint (Nm)	NAVDEC Trajectory to the Next Waypoint (Nm)	Difference (%)
1	328	20	15.03	32.80	118
2	288.3	35	18.57	38.14	105
3	324.9	48	15.03	39.07	160
4	343	338.5	15.07	15.07	0
5	338.8	338.8	15.09	15.58	3
6	324.8	326	15	15	0
7	325.1	327	15.14	15	‒1
8	339	337.5	15.22	15.36	1
9	336.6	336.6	15.1	15.23	1
10	340.4	340.4	15.03	15.09	0.4
11	150.8	290	15.02	27.46	83
12	147.7	147.7	15.01	15.1	0.6
13	147.3	147.3	15.01	15.07	0.4
14	142.3	125	15.02	16.85	12

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
