# Peer review of "Beam Search Algorithm for Ship Anti-Collision Trajectory Planning"

_sensors, 2019, doi:10.3390/s19245338_

Round 1
Reviewer 1 Report
Generally, this paper is interesting and presents results which can be accepted. To be more promising, latest works on path/trajectory planning and collision avoidance for surface vehicles (e.g., Ocean Eng., 2019, 184: 1-10; J. Navigation, 2013, 66(2): 265-281) should be intensively reviewed. Furthermore, a pseudo-code algorithm is highly recommended to summarize this work.
Author Response
English language and style
( ) Extensive editing of English language and style required
(x) Moderate English changes required
( ) English language and style are fine/minor spell check required
( ) I don't feel qualified to judge about the English language and style
The proofreading of the article was done.
Generally, this paper is interesting and presents results which can be accepted. To be more promising, latest works on path/trajectory planning and collision avoidance for surface vehicles (e.g., Ocean Eng., 2019, 184: 1-10; J. Navigation, 2013, 66(2): 265-281) should be intensively reviewed.
In [19] the fast marching method is used in multilayer path planner for an unmanned surface vehicle. This planner takes into account complex marine environments with coastal and dynamic obstacles. In [17] and [20] the artificial potential field method is presented as a tool to solve local optimization of path planning and to avoid dynamic obstacles. It can be noticed that ship domains are important approaches to dealing with the optimal path planning problem [21-22].
Furthermore, a pseudo-code algorithm is highly recommended to summarize this work.
The pseudocode of the algorithm is given on page 5. Duo to algorithm complexness the authors decided it’s a better form or presentation than a flowchart.
The pseudocode of the algorithm is given below:
CT- list of current trajectories (T in CT)
FT- set of final trajectories
OT- own trajectory
asT- an current processed segment of T trajectory
maxN - maximal number of processed solutions
Add OT to CT
While CT is not empty
T - first trajectory from CT
Remove T from the start of list CT
For each Target_i ship's trajectory
If DCPA of OWN is violated in section asT by Target_i trajectory
generate two anti-collision manoeuvres (left and right) and segments back to the waypoint
Based on the two manoeuvres and T, create two new trajectories T1 and T2
Else Generate artificial collision course of OWN ship with Target_i trajectory.
For the artificial course, generate two anti-collision manoeuvres port (and starboard) and segments back to the waypoint
Based on the two manoeuvres and T create two new trajectories T1 and T2
For each trajectory from {T1, T2} set
If the first of two newly created segments of Ti trajectory is DCPA safe with respect to all remaining ships
If segments are last segments in Ti and both are DCPA safe
Add Ti to FT
Else
Add Ti to the end of list CT
If size(CT) > maxN
Remove size(CT)-maxN worst trajectories (in terms of expected length) from CT
Choose the best trajectory (or trajectories) from FT.
Reviewer 2 Report
An interesting paper for collision avoidance however there are some limitations as follows:
Lines 43-44: there is a discussion about the importance of consumption which is not well supported in the paper.
Lines 78-79: COLREGS should not be limited or excused by adverse weather. Any evidence to the contrary should be referenced by authors.
Line 157: A pseudo-code is presented. However a skilful navigator should be able to graphically solve similar triangle for 3 targets in less than 10 minutes. Therefore the benefits from proposed code should be supported.
Furthermore, the assumptions of the code should include a safe margin which excludes weather, current and equipment usual errors such as Gyro compass, AIS, ARPA
Line 38: Alterations of a ship are limited only to the right. This should be included in the paper.
Line 241: a useful discussion would be how a speed reduction could affect the manoeuvring and saving of fuels.
Line 273: Crowded situation should be discussed if other ships followed COLREGS which usually is not in such cases.
Line 297: the saving of 0.13 nautical miles are important only for the current experiment and cannot be generalised. For instance, several ships sail for thousands of miles and such distance is insignificant.
Line 310: An improvement of 37% in such a short voyage is significate but could be a unique case. A comparison with stoppages or speed reduction should be included. The presentation of ship characteristics would be useful for this case.
Line 319: Rule 2b of COLREG is misrepresented. The rule actually has validity that only in the case of great risk of collision where alteration to right side and speed reduction can not stop the collision which is certain.
Author Response
Lines 43-44: there is a discussion about the importance of consumption which is not well supported in the paper.
Below part added in the discussion section. In Author's opinion, it gives a clear example, how much money can be saved and how much we can reduce a carbon footprint using collision avoidance systems.
Referring the results to the container ship presented in the introduction, it can save over 800 tons of fuel every year (14 collision avoidance manoeuvres per day). The average global price of one ton of heavy fuel oil on 13.11.2019 was 406.50 USD, which translates into saving over 300,000 USD, and much more from the environmental protection perspective. This saving can be almost doubled, when ships use marine gas oil, which is already required in certain areas around the world.
Lines 78-79: COLREGS should not be limited or excused by adverse weather. Any evidence to the contrary should be referenced by authors.
“bad weather conditions most of the vessels do not fulfil “safe speed” requirements as it results in huge delays and destroying ports and vessels schedules”
The bad weather condition in Authors opinion is also restricted visibility. The long personal experience of the author, who spent many years at sea, shows, that most of the vessels do not reduce speed due to for example fog. Otherwise, as it is mentioned in the article, it results in huge delays and destroying ports and vessels schedules.
The author changed misleading “bad weather: into “restricted visibility”
Line 157: A pseudo-code is presented. However a skillful navigator should be able to graphically solve similar triangle for 3 targets in less than 10 minutes. Therefore the benefits from proposed code should be supported.
The maximum number of vessels which can be analysed by BSA is equal to 30, which is much more than 3. Such information was added after the pseudocode.
Furthermore, the assumptions of the code should include a safe margin which excludes weather, current and equipment usual errors such as Gyro compass, AIS, ARPA
Authors mentioned in the assumption part, that DCPA was set up to 1 Nm. The main source of information for NAVDEC is AIS. Taking into account these two facts, even the standard errors of GPS will still permit to pass each other on the safety distance. In the case of other errors, the user will be informed by the interface, that GPS should not be used to estimate ship position.
Line 38: Alterations of a ship are limited only to the right. This should be included in the paper.
NAVDEC fully follows COLREGs, this is why has the priority to alter course to starboard. It was added to the article.
Line 241: a useful discussion would be how a speed reduction could affect the manoeuvring and saving of fuels.
In all cases, the NAVDEC was observing the COLREGs, which means that it always suggested an alteration to starboard. Some of these alterations had to be significant to fulfil the Regulations. It was proven by the authors (fig. 5) that in some cases, a small alteration to port could result in a significant reduction of the distance, particularly when there are many ships around. The case in Fig. 5, can be considered as a heavy traffic situation in the Baltic Sea. Within six Nm from Wolin, there were seven other vessels. At least one of them on the opposite course, which made a speed reduction a non-effective manoeuvre. In this case, speed reduction could increase only TCPA, while DCPA remains the same when the other vessel keeps her course and speed. Speed reduction is an effective manoeuvre in the crossing situation and in restricted waters when course alteration is limited, e.g. by the width of the channel. Such manoeuvre reduces fuel consumption. However, in the open sea course alteration is used much more often than speed reduction. Such instruction is issued by Masters in ‘standing orders’. Course alteration, where possible, is considered as faster and well visible by other navigators.
Discussion included in the paper, section Results, before Table 1.
Line 273: Crowded situation should be discussed if other ships followed COLREGS which usually is not in such cases.
In all cases, the NAVDEC was observing the COLREGs, which means that it always suggested an alteration to starboard. Some of these alterations had to be significant to fulfil the Regulations. It was proven by the authors (fig. 5) that in some cases, a small alteration to port could result in a significant reduction of the distance, particularly when there are many ships around. The case in Fig. 5, can be considered as a heavy traffic situation in the Baltic Sea. Within six Nm from Wolin, there were seven other vessels.
Discussion included in the paper, section Results, before Table 1.
Line 297: the saving of 0.13 nautical miles are important only for the current experiment and cannot be generalised. For instance, several ships sail for thousands of miles and such distance is insignificant.
Authors presented all results collected during the voyage, including those, which reduced the distance by 0,13 Nm and even that, which made the route longer. Authors agree with the Reviewer, that 0,13 Nm is not the significant result, however, 37% is already significant.
Line 310: An improvement of 37% in such a short voyage is significate but could be a unique case. A comparison with stoppages or speed reduction should be included. The presentation of ship characteristics would be useful for this case.
Authors presented the results of implementation on ferry Wolin and are fully aware that could be a unique case. However, it is the real case. In the Author's opinion significant. Reviewer opinion in this field is highly appreciated. Authors undergoing talks with shipowner to carry out a test on a seagoing vessel. That will deliver more results including stoppages and speed reduction. However, ferry characteristics and very tight schedule did not allow Authors to use the engine for collision avoidance manoeuvre.
Line 319: Rule 2b of COLREG is misrepresented. The rule actually has validity that only in the case of great risk of collision where alteration to right side and speed reduction can not stop the collision which is certain.
Authors fully agree with Reviewer interpretation. The full interpretation was included in the article. Rule 2b is just an example of challenges for autonomous shipping.
Reviewer 3 Report
The authors present an interesting approach to determine energy-efficient paths for collision avoidance. The paper is well structured and the need is highlighted. However, I have some concerns regarding the article and its application.
Firstly the language should be proofread and improved. The article will benefit from clear and precise language.
Purpose and discussion:
What are the main factors affecting fuel consumption? How does your article fit into these? For example, How would slow steaming affect safety and fuel consumption? What about overall route optimization in comparison to trajectory planning?
Is weather-optimized routing, not a more efficient tool for the overall voyage plan, to save fuel? How much do you expect to save in comparison to such an approach? Or at least compare the benefits.
The planning and calculation of the algorithm were mentioned as being 2 seconds, could you provide a reference value on how much time will be available to avoid a collision?
In the discussion:
How will the results be affected if the COLREGs are considered? Will the calculated route be different from the NAVDEC route? What leads you to the claim that the routes are safe when they violate the COLREGS and be potentially misinterpreted.
How do your assumptions that are presented in sections 2 and 3 (Which are very well presented) affect your results?
Simulations presented in this article or further work should be carried out that include a reactive target ship model to test this?
Minor:
Could you present more information on the NAVDEC (producer, settings, etc.) earlier and in more detail.
Round 2
Reviewer 2 Report
Thanks to the authors for significant improvement. Still some concerns from seamanship practices however it is a useful tool presented.
Author Response
Thanks to the authors for significant improvement. Still some concerns from seamanship practices however it is a useful tool presented.
Thank you very much for your comments and inputs. We will pay much more attention to seamanship practices in our following articles.
Reviewer 3 Report
Dear authors,
The manuscript has improved substantially.
Some minor comments that are left:
Page 2, lines 63 -70 insert should be presented later, in the discussion
Line 91, relevant research was carried out and summarized by Brekke et al. 2019, https://iopscience.iop.org/article/10.1088/1742-6596/1357/1/012020
Especially the Branching course MPC shares some similarities with the research presented here. Could you add in the discussion a comparison/ brief discussion of the differences/ improvements from that algorithm?
In figure 4, if the algorithm would consider Colregs, the ship number C would need to take avoiding action. Were such situations encountered in the case study?
Author Response
Page 2, lines 63 -70 insert should be presented later, in the discussion
Above mentioned lines are now located in the discussion, the second paragraph.
Line 91, relevant research was carried out and summarized by Brekke et al. 2019, https://iopscience.iop.org/article/10.1088/1742-6596/1357/1/012020
Especially the Branching course MPC shares some similarities with the research presented here. Could you add in the discussion a comparison/ brief discussion of the differences/ improvements from that algorithm?
Branching course MPC algorithm considers a different aspect of trajectory generation. It incorporates ship dynamics to find the solution based on anti-collision trajectory given as input (determined by another algorithm). On the other hand, our Beam Search method generates an anti-collision trajectory from scratch (earlier stage of trajectory planning than Branching course MPC). We are still working on incorporating ship dynamics into our solution.
In conclusions we added references to the mentioned research (when discussing incorporating ship dynamics in our algorithm):
[36] Eriksen, B. O. H.; Breivik, M.; Wilthil, E .F.; Brekke, E. F. 2019 The branching-course MPC algorithm for maritime collision avoidance. J. Field Robotics 2019, DOI:10.1002/rob.21900
[37] Brekke,E. F.; Wilthil, E. F.; Eriksen, B-O. H.; Kufoalor, D. K. M; Helgesen, K.; Hagen, I. B.; Breivik, M.; Johansen T. A. The H.; project: Developing closed-loop target tracking and collision avoidance systems. Journal of Physics: Conference Series, Volume 1357(138), 2019, DOI: 10.1088/1742-6596/1357/1/012020
In figure 4, if the algorithm would consider Colregs, the ship number C would need to take avoiding action. Were such situations encountered in the case study?
In figure 6, there is a situation with the vessel Najland approaching from port side. In general the situation is not similar to that in figure 4, however, “behaviour” of approaching vessels were quite similar. Both i.e. Najland and number C did not take avoiding action when they noticed, that stand on vessel already alter her course to avoid the collision. It is rather usual action in complex situation, i.e. when more than two vessels are involved. In figure 4 ship A is give way vessel in relation to ship B, while it is stand on vessel in relation to ship C. In figure 6, Wolin is give way vessel in relation to MMSI 245170000, while stand on vessel in relation to Najland. Collision avoiding trajectory calculated by BSA solved both challenges i.e. collision situation with MMSI 245170000 and with Najland. When Najland noticed such action executed by Wolin, did not take any action despite she was give way vessel.